# The Impact of AI-Recommended Content Affordances on Post-Purchase Intention in Stockout Substitution Scenarios

**DOI:** 10.3390/bs15111507

**Published:** 2025-11-06

**Authors:** Beibei Dai, Jianming Zhu, Xiaoling Zhu

**Affiliations:** School of Information, Central University of Finance and Economics, Beijing 102206, China; dbb@email.cufe.edu.cn (B.D.); 2021110187@email.cufe.edu.cn (X.Z.)

**Keywords:** AI recommendations, substitution strategies, technology affordance, perceived value, privacy concerns, post-purchase intention

## Abstract

Stockouts significantly threaten consumer loyalty and cause substantial economic losses. In response, online platforms are widely deploying AI recommender systems to provide substitutes. However, whether such AI-driven substitution strategies can effectively mitigate the negative consequences of stockouts remains underexplored. Grounded in technology affordance and perceived value theories, this study develops a conceptual framework to investigate how content affordances of AI-recommended substitutes—specifically perceived fit, personalization, and serendipity—influence post-purchase intentions through functional and emotional value perceptions. Analysis of survey data from 479 respondents reveals that these affordances enhance perceived value, which in turn strengthens post-purchase intentions. Moreover, the findings demonstrate distinct effects of each affordance dimension on perceived functional value versus emotional value. In terms of the moderating effects, privacy concerns positively moderate the relationship between perceived functional value and post-purchase intention. Necessary Condition Analysis (NCA) further identifies critical prerequisites for achieving high perceived value and post-purchase intentions. This study extends the application of AI recommender systems to service recovery contexts and offers a wealth of novel insights for designing effective substitution strategies.

## 1. Introduction

A stockout is defined as a state of product unavailability or a temporary mismatch between supply and demand for inventory items ([39]). It is a service failure phenomenon that may elicit strong negative consumer reactions. A survey on online shopping annoyances revealed that stockouts are the second most prevalent issue ([10]). Therefore, developing effective stockout recovery processes has become a critical challenge in online retail management. Product substitution has been proven as a key strategy to mitigate negative effects ([38]). Recently, AI-recommended agents have emerged as a novel tool for service recovery. In contrast to conventional marketers offering generic alternatives, AI systems generate a unique set of substitutes tailored to heterogeneous individuals. Existing studies have demonstrated that AI recommender systems leverage robust learning, adaptation, and interactive technical capabilities to construct and refine hyper-personalized customer experiences, thereby enhancing click-through rates, value co-creation, and engagement behaviors.

Despite the extensive deployment of AI recommender systems, their efficacy as a recovery strategy for stockouts lacks robust theoretical grounding. Prior literature exhibits dual limitations in explaining this phenomenon. First, studies predominantly focus on traditional offline retail, lacking validation of the efficacy of AI agents as digital remedial tools. Recommender systems harness advanced algorithms to generate assortment sets that surpass traditional marketing approaches by providing unprecedented insight into and response to consumer behavior ([92]). The AI-recommended items are shaped by the interplay of individual preferences, task context, and technical features ([16]). User preferences derived from both explicit data and implicit feedback are used to select substitutes that exhibit congruence with consumers’ tastes, colors, brands, or historical consumption experiences ([80]). Our research aligns with another unresolved gap in the current literature: existing research emphasizes attribute fit between substitutes and out-of-stock products while overlooking the critical role of user-specific preference matching. In fact, consumers’ choice behaviors are driven by personalized factors including idiosyncratic preferences, individual values, and habitual behaviors ([5]).

Second, the efficacy of substitution strategies constitutes a pivotal research direction in substitution studies ([5]). Current research predominantly examines the impact of stockouts on consumers’ single purchase decisions through fine-grained analyses of substitutive attribute spaces, including categories, brands, and functionality, to pinpoint optimal substitutes for customer retention ([44]).Yet, it rarely evaluates the long-term effects of substitutive recovery strategies, especially in pivotal strategic dimensions such as relationship rehabilitation and loyalty cultivation, while short-term profit maximization may undermine sustainable consumer value ([70]). Therefore, consumer satisfaction recovery and loyalty restoration are pivotal components in service recovery frameworks ([110]). Improper service recovery interventions will amplify initial service failure effects; conversely, the service recovery paradox holds that effective recovery strategies can compensate for service failures and convert dissatisfied customers into loyal advocates ([75]). In particular, a 5% improvement in customer retention yields 25–85% profit growth ([85]).

To fill this gap, this research establishes a theoretical framework that integrates technological affordance theory and perceived value theory, conceptualizing the content affordances of AI recommendations as strategic recovery resources. The study provides pioneering insights into how AI-driven functional designs produce long-term recovery effects in out-of-stock substitution scenarios. Post-purchase intentions have been used to forecast consumers’ long-term economic and social behaviors in terms of consumers’ repurchasing intentions and word-of-mouth (WOM) following service recovery ([71]). Yet, it is rarely used to evaluate the effectiveness of substitution strategies. In contrast to the short-term purchasing decisions, post-purchase intentions offer an effective way to assess the effect of service recovery strategies on enhancing customer experience and relationship and converting potential losses into opportunities to boost customer loyalty and satisfaction ([81]). Perceived value has been proven to be a critical antecedent of post- purchase intentions ([49]). As demonstrated in existing literature, the concept of perceived value is multidimensional. The value dimensions of utilitarian and hedonic encompass a wide range of specific types of values, describing consumers’ perceptions on the preset value proposition of service providers during interaction ([58]).

Meanwhile, affordance is a popular perspective in the field of information systems to examine digital technology innovation and its efficacy potential. It describes the relationship between “value in use” and the technical features that provide potential for specific goals ([55]). Accordingly, as a content information service based on preset programs, the realized value of content design for AI-recommended alternatives hinges on consumers’ perceptions of the technology’s usefulness and how it can be used to achieve goals. Based on this discussion, this study delineates the causal pathway from the content affordances of AI recommender systems to post-purchase intentions through perceived value. Furthermore, distinct technological features exert varying impacts on consumer cognition and affect ([36]), hence, this study explores the differential effects of content affordances on perceived functional value and perceived emotional value. Additionally, we employ Necessary Condition Analysis (NCA) to reveal the necessary conditions for producing the expected effects to improve diagnostic ability.

In addition, the underlying mechanism of recommender systems inherently induces privacy concerns ([98]). Individuals’ perceptions of privacy disclosure risks or potential negative consequences significantly influence users’ decisions to adopt and continue using AI-driven services ([14]). Current research demonstrates no consensus on the impact of privacy concerns on service adoption: Many studies suggest that privacy concerns inhibit consumer engagement with AI services, but paradoxically, others indicate a positive effect ([6]). Although previous research acknowledges that privacy holds utilitarian value as foundational to human autonomy ([41]), the crucial role of privacy concerns in shaping consumers’ platform switching behavior remains unexamined. Given that individuals with high and low levels of privacy concerns represent differences in cost–benefit tradeoffs ([52]), this study further discusses the moderating role of privacy concerns. The main findings are highly significant for platform retailers managing product stockouts in the presence of substitutes, contributing to improved product sales and inventory management policies.

The remainder of this paper is structured as follows. First, we provide a review of substitution for out-of-stock, technology affordance and perceived value theory. Second, we present a theoretical framework and a set of testable hypotheses. Third, we detail the methods and data analysis results. Finally, we discuss the findings, implications, limits, and future research directions.

## 2. Theoretical Background

### 2.1. Substitution for Out-of-Stock

Consumer reactions to substitution policies for stockouts vary widely, with policies implemented by either the retailer or the customer ([106]). Research on customer-driven product substitution explores intrinsic motivations, choice preferences, and product attributes in alternative selection. Contextual awareness and heuristic processing of market cues shape consumers’ adoption of substitutes for brands, price, categories or platforms. For instance, mismatch costs may lead individuals to seek substitutes within the same platform, whereas monetary or temporal opportunity costs induce platform switching behavior ([23]). Moreover, consumers attribute stockouts to either high demand or limited supply, inferring the product’s popularity or scarcity ([26]). To minimize anticipated regret, they subsequently increase purchases of similar products. However, when consumers regard retailers’ substitution offers as a restriction on freedom of choice, the boomerang effect triggers psychological reactance, resulting in dissimilar purchases or decision deferral ([29]).

Substitution strategies for stockouts initiated by retailers are supply side solutions. Extant studies focus on the configuration of substitute sets for inventory optimization, as the attribute of choice sets significantly influence consumer-driven choice. Moreover, the situational effect literature is an important stream of relevant research. A new item that is introduced alters the composition and dominance structure of existing choice sets. Consumers tend to evaluate substitutes based on critical product attributes, price levels, and spatial positions of phantom options. This evaluation process leads to consumers’ choice behavior that deviates from standard rational models ([23]), resulting in disproportionate preference shifts manifested as the similarity effect, attraction effect, or compromise effect. Potential explanations include loss aversion, attribute importance shifts, and similarity heuristics, motivating the use of task simplification strategies to reduce decision errors or anticipated regret ([23]). Beyond attribute value comparisons, attribute (dis)similarity cues within identical set architectures intuitively direct consumer attention to divergent options. However, under risk conditions or in analytical processing modes, consumers exhibit herd behavior toward products with similar attributes.

AI recommender systems serve as retailer-driven, differentiated stockout tools in the digital intelligence era ([109]). Despite their widespread adoption in retail, little research has examined their efficacy in stockout recovery. Limited studies cover the decoy effect, inventory management, or pricing strategy in the context of recommender systems. For example, [74] ([74]) demonstrated that including a decoy in personalized recommendations minimizes demand for a target option, deviating from the traditional decoy effect. [84] ([84]) proposed a mixed-integer programming algorithm to allocate recommended items among buyers while accounting for individual preferences and real-time inventory states, including uncertainty. [106] ([106]) argued that investing in high-quality recommender systems can reduce search costs for unfamiliar substitutes but requires cost–benefit trade-offs.

Most existing studies compare the attributes of substitute products and stockouts to test the substitution effect or focus on the utility of recommender systems in inventory management. However, few studies have investigated how the technology affordance of recommender systems influence the effectiveness of alternative strategies. In particular, the content generated by AI recommender systems not only represents task consistency through functional or attribute substitution but also reflects self-consistency aligned with user preferences, needs or personality, thereby manifesting the algorithmic logic and technical features of AI recommender systems. These factors are crucial for predicting information adoption and recommendation acceptance ([104]). Thus, this study integrates technology affordance theory and perceived value theory to investigate how the content characteristics of choice sets of recommender systems transcend functional substitution, thereby enhancing service recovery effectiveness. Rather than regarding recommender systems as integrated tools to advance prior work, our framework dissects their content attributes, thus revealing how algorithmic content design affects satisfaction and loyalty in service recovery following a stockout. It also provides novel insights into the application of AI recommender systems in the stockout substitution field.

### 2.2. Technology Affordance

Technology affordance, which offers a perspective of social materiality, illuminates the relationship between users and technology within a situated environment ([18]). It comprises two key dimensions: technical functionality and goal-oriented action ([56]). Technological usability hinges on human perceptions of the functions offered by a technological object, suggesting that technological capabilities are part of the relationship rather than inherent attributes ([102]). Moreover, the realization of technology affordance requires goal-oriented consumers to trigger technical operations, with their perceptions and interpretations of technological features evolving after each action ([40]), following a key process sequence of technology affordance, belief formation, and behavioral outcomes. [102] ([102]) proposed that perceived affordance is the possibility for users to infer specific actions via technical features and functions, and this plays an important role in understanding the human–computer interaction process. Therefore, this study investigates technology affordance from the perspective of perceptual artifact design related to content algorithm—that is, consumers’ perception of the potential for action embedded in the content attributes of the choice set.

In stockout situations, consumers’ decision frames center on specific attributes such as brand and flavor ([12]), leading to a preference for substitutes that closely resemble their original choice ([5]). The perceived fit between product attributes serves as a functional cue in consumers’ heuristic decision-making process. Therefore, functional substitution and utility restoration are established as the first dimension of content affordance influencing consumers’ service recovery experience. Drawing on relevant research in fields such as brand similarity and search similarity, this study adopts perceived fit to describe the overall compatibility between substitutes and out-of-stock products. It is defined as “the overlap of attributes or associations between substitutes and target products” ([53]). Furthermore, personalized affordances manifest functional attributes inherent in content ([56]). The items filtered by the AI recommender system are closely congruent with consumers’ preferences, reflective of their self-image, and thus critical in shaping consumer choices and establishing brand loyalty ([89]). [57] ([57]) proposed that personalization needs to be noticed to be effective. Thus, perceived personalization is the second factor of functional experiences, defined as “consumers’ subjective perception of the extent to which service providers offer personalized services” ([93]).

Perceived serendipity represents an algorithmic affordance ([107]). Prior literature indicates that although preference prediction accuracy is the most important performance criterion for recommender systems, failure to discover users’ emerging interests may gradually generate filter bubbles ([24]), thereby compromising the long-term efficacy of recommendations. Researchers strive to strike a balance among accuracy, diversity, novelty, and serendipity ([43]). Thus, perceived serendipity is regarded as the third content affordance dimension, related to the content attributes of relevance and novelty ([63]). Moreover, relevance is one of the key differences between diversity and serendipity. Diversity does not include relevance, whereas relevance is vital for serendipity ([47]). In out-of-stock situations, substitute set attributes correlate with specific decision goals for alternative selections. Thus, perceived serendipity is a key content feature of the decision scenario, combining informational value and surprise, defined as “consumers’ perceived extent to which recommended substitutes reveal value beyond initial expectations” ([105]).

### 2.3. Perceived Value Theory

Perceived value is a crucial determinant of consumer shopping experiences. It is defined as “consumers’ overall assessment of a product’s (or service’s) utility based on perceptions of what is received and what is given” ([46]). Previous research acknowledges perceived value as a key explanatory factor of consumer satisfaction, purchase decisions, and loyalty behaviors. Treating it as a multidimensional construct rather than a unitary construct better clarifies its antecedents and consequences. [97] ([97]) suggest that technology enables both utilitarian values and hedonic values, with algorithms exerting dual cognitive and affective influences on consumers ([103]). Building on this dual influence and drawing on the research findings of [87] ([87]) on perceived value, this study investigates consumer responses to AI-recommended substitutes through two dimensions of perceived functional and perceived emotional value. Perceived functional value represents consumers’ rational cost–benefit assessment of choice utility, defined as “the perceived utility derived from functional, utilitarian, or physical attributes of AI-recommended substitutes”. Perceived emotional value, on the other hand, is defined as “the perceived utility obtained from the ability of AI-recommended substitutes to arouse emotions or emotional states” ([46]), such as pleasure, surprise, and enjoyment.

## 3. Research Hypothesis and Theoretical Framework

### 3.1. Antecedents: Drivers of Perceived Value

#### 3.1.1. Functional Pathways

Stockouts increase consumers’ attention to substitutes that share important attributes with their preferred products. Categorization theory posits that consumers classify objects into distinct categories to enhance their comprehension of the environment. Thus, when a product matches the attributes of a category member, cognitive schema for the preferred product activates the evaluation of related products ([27]). In stockout scenarios, consumers prefer similar substitutes to infer the common attributes, reducing purchase uncertainty and mis-selection risks. The technical capability of AI recommender systems to suggest similar products aligns with task demands, enhancing consumers’ perceptions of the usefulness of new technologies ([16]). Meanwhile, by actively recommending similar substitutes, AI recommender systems can reduce consumers’ search time and cognitive load, as well as enhance the economic utility of alternative choices through the functional benefits of perceived fit. In addition, perceived fit is a task-oriented content affordance dimension that provides key functional cues, and information accuracy is considered to be an important dimension of e-service quality ([50]), which is related to perceived functional value. Thus, we hypothesize that:

**H1.** 
*Perceived fit positively influences consumers’ perceived functional value.*


Perceived personalized substitutes significantly enhance consumers’ perception of functional value. AI recommender agents track users’ online behaviors and dynamically adapt displayed results through interactions, providing personalized substitutes with higher information quality and accuracy. This process effectively reduces both users’ search costs and information overload ([83]). Furthermore, perceived personalization fosters cognitive trust in recommender agents. Compared to generic alternatives, personalized functions better emulate customer decision patterns ([4]). Consequently, personalized substitutes lead consumers to believe that AI recommendations can enhance the effectiveness of their chosen alternatives. Moreover, personalized products reflect personal interests, preferences, values, and personality traits, promoting consumers’ self-referential processing through material possession ([7]). Thus, these customized substitutes serve as critical symbolic vehicles for consumers to express their unique identities. Substitutes function as symbols that overlap with self-concept and aid consumers in their pursuit of a coherent self-image, while the functional value is tied to the underlying needs for social recognition or personal expression. Thus, we hypothesize that:

**H2.** 
*Perceived personalization positively influences consumers’ perceived functional value.*


Serendipity is interpreted as “unexpected discoveries and information” ([104]). Perceived serendipitous items may help consumers realize new solutions, ideas, or other unexpected directions. For instance, AI recommender agents not only provide laundry detergents with similar attributes but also help consumers discover alternatives, such as liquid detergents or laundry pods, which meet their specific cleaning needs. [104] ([104]) revealed that the additional information gained from serendipitous content helps users effectively evaluate decision options, enabling them to better understand products and compare potential performance. Furthermore, [64] ([64]) found that serendipitous information discovery surpasses consumers’ expectations while fulfilling their personal information needs. Additionally, diversity and novelty are critical determinants of serendipity. They expand consumers’ interests, tastes, and perspectives while mitigating the excessive specialization issues caused by “filter bubbles” or “echo chambers,” thereby improving the quality of alternative purchase decisions. According to [104] ([104]), serendipitous content enhances the perceived usefulness of recommendations. Thus, we hypothesize that:

**H3.** 
*Perceived serendipity positively influences consumers’ perceived functional value.*


#### 3.1.2. Emotional Pathways

Stockouts may induce consumer anxiety and aversion due to a loss of control ([77]). Compensatory control theory holds that humans have a strong motivation to seek control restoration. Reestablishing and restoring control over product choices serves as an effective means to compensate for the absence of control in a disordered environment ([72]). In stockout scenarios, perceived-fit substitutes offer functional attributes similar to preferred products and improve the efficacy of repurchase decisions during information overload. [78] ([78]) demonstrate that experiences of efficacy and fulfillment act as motivators for psychological ownership formation. Notably, psychological ownership formation elevates consumers’ sense of control ([31]). Thus, perceived-fit substitutes can be viewed as an alternative compensatory form for objectively restoring control. [69] ([69]) also proposed that threats to personal control amplify the attractiveness of similarity. The fulfillment of psychological needs mitigates tension and anxiety induced by a loss of control, thereby deriving enjoyment that aids in psychological recovery. Thus, we hypothesize that:

**H4.** 
*Perceived fit positively influences consumers’ perceived emotional value.*


Perceived personalized substitutes can evoke users’ positive emotional experiences. AI technology enhances recommender systems’ external data learning capacity and personalization quality. High-quality personalized substitutes increase consumers’ sense of agency and control, leading to a relaxed feeling related to self-efficacy ([100]). Furthermore, personalized substitutes more effectively attract consumer attention and trigger positive cognitions responses to the content. [51] ([51]) indicated that when content is perceived as personalized, users experience a higher degree of enjoyment and content consistency. In addition, personalized information strengthens consumers’ perception of the similarity between themselves and product or brand images ([93]). Users prefer and show positive effects toward self-relevant information, as self-concept consistency brings individuals higher hedonic enjoyment and well-being ([20]). Thus, we hypothesize that:

**H5.** 
*Perceived personalization positively influences consumers’ perceived emotional value.*


Perceived serendipity describes an “Aha!” moment in which affective dimensions induce pleasurable consumer experiences. The serendipitous discovery mechanism is intrinsically linked to expectation violation ([32]). When unpredictable product attributes violate consumer expectations, the resulting information asymmetry creates dissonance between pre-existing mental schemata and newly formed beliefs after recommendation experiences. This means that consumers who browse serendipitous alternatives are more likely to evoke feelings of “surprise and joy” without active searching ([11]), and this strong emotion will amplify users’ positive attitude towards cognition. Moreover, while substitutes matching out-of-stock products may overexpose consumers to homogeneous attributes, serendipitous alternatives differ from the anchor product on specific dimensions, redirecting attention toward unexpectedly relevant discoveries. Novel knowledge and information activate consumer inspiration ([96]), generating surprise and emotional fulfillment. Thus, we hypothesize that:

**H6.** 
*Perceived serendipity positively influences consumers’ perceived emotional value.*


### 3.2. Outcomes: Impact on Post-Purchase Intention

Post-purchase intention refers to the customer’s intention to repurchase products or services from the same retailer and share their usage experience with friends or others ([71]). [9] ([9]) proposed that repurchase intention and WOM can be used to evaluate post-purchase behavior. Consumers’ perceived value of AI-generated recommendations during out-of-stock episodes exerts a positive effect on post-purchase intentions. On the one hand, consumers expect high-quality functional recovery services, with AI-recommended substitutes providing convenience, problem-solving capability, and professional service quality as functional utilities. Consumers’ assessments of service recovery value manifest in their behavioral intentions, which drive continued consumption and facilitate positive WOM dissemination. [65] ([65]) demonstrate that repurchase behavior of experienced users is rooted in the relationship between consumers and platforms, with perceived value constituting the core foundation of relational exchange. On the other hand, the emotional value perceived by consumers in platform content fosters commitment and loyalty toward the platform enterprise ([86]). Pleasant or surprising psychological experiences, coupled with psychological compensation for the loss of control, elevate consumers’ service recovery satisfaction. According to the service recovery paradox, properly executed recovery strategies can enhance consumers’ positive WOM and repurchase intention. Thus, we hypothesize that:

**H7a.** 
*Perceived functional value positively influences consumers’ post-purchase intentions.*


**H7b.** 
*Perceived emotional value positively influences consumers’ post-purchase intentions.*


### 3.3. Comparing Functional and Emotional Value

Different dimensions of content affordances in AI-recommended substitutes exert varying degrees of influence on users’ perceived functional value and emotional value. Functional value originates from goal orientation, rationality, and utilitarian purposes during product/service usage, being more closely associated with individual cognition ([37]). Emotional value exhibits greater subjectivity and encompasses hedonic emotional traits such as interest, joy, and entertainment ([76]). [3] ([3]) confirmed that utilitarian attributes of applications exert stronger effects on consumers’ cognitive responses than affective responses, whereas hedonic attributes predominantly influence affective reactions rather than cognitive evaluations. [54] ([54]) proposed that consumers’ value perception is contingent upon their goals and situations.

Perceived fit captures the similarity transfer between stockout items and substitutes, emphasizing content affordance design for substitution tasks and functional remediation. Compared to emotional value, it more strongly influences users’ evaluations of service recovery effectiveness and performance. The direct result of perceived personalization is to evoke consumers’ awareness of the consistency between substitutes, personal interests, and self-concepts ([95]), while self-consistency is more related to affect-based structures ([2]). [57] ([57]) proposed that actual personalization elements to be effective must be perceived. Considering users as individuals with unique needs is a crucial prerequisite for achieving positive personalized outcomes ([83]). Unlike economically oriented functional value, consumers’ need for uniqueness fulfillment exhibits stronger associations with the emotional value of sensory pleasure and self-enhancement, with personalized information experience being a pleasurable response state. Perceived serendipity encompasses the dual concepts of information value and unexpectedness. The “Aha moment” triggered by unexpected information that exceeds initial expectations activates or amplifies users’ pleasant emotions, while the surprise affect itself originates from algorithmic content design that expands user preferences or incorporates “surprise” options ([47]). Thus, we hypothesize that:

**H8a.** 
*Perceived fit has a stronger positive effect on perceived functional value than perceived emotional value.*


**H8b.** 
*Perceived personalization has a stronger positive effect on perceived emotional value than perceived functional value.*


**H8c.** 
*The effects of perceived serendipity on perceived functional value and perceived emotional value do not differ significantly.*


### 3.4. Boundary Conditions

Privacy concerns refer to an individual’s level of worries and discomfort related to potential platform privacy violations, such as unauthorized disclosure of information and non-consensual secondary use ([42]). [62] ([62]) proposed that privacy is a utilitarian tool, and privacy concerns act as cost factors in the exchange of interests and the establishment of special relationships between consumers and retailers. [66] ([66]) further posit that these concerns reflect efforts to mitigate disclosure risk. Individuals’ risk beliefs lead to varying perceptions of privacy threat severity ([8]). Compared to individuals with low privacy concerns, those with high levels possess heightened sensitivity regarding their personal data ([79]) and tend to employ prevention-focused behavioral strategies ([66]), such as defensive or avoidance behaviors ([68]), thereby reducing information disclosure and mitigating potential negative consequences.

Social exchange theory posits that individual social behaviors are a series of interdependent exchange processes, where participants seek to maximize benefits and minimize costs ([1]). In out-of-stock substitution scenarios, switching to a new platform requires consumers to re-disclose detailed self-relevant information, including browsing habits, purchasing behaviors, and social interactions. This process exacerbates their privacy concerns and psychological costs, establishing high initial perceived switching costs that may trigger negative responses such as avoidance behaviors ([34]), negative attitudes ([73]), and psychological discomfort ([30]). However, if the original platform can promptly provide service recovery measures with high perceived value, the benefits users gain from utilitarian, hedonic, and emotional rewards not only enhance their trust and loyalty toward the original platform but also raise the psychological and behavioral barriers to leaving the existing relationship ([15]). Furthermore, Wang’s research confirms that high perceived switching costs promote customer loyalty to the original service providers ([99]), with this effect being more pronounced among consumers who hold stronger beliefs about privacy threat severity ([67]). Through this cost–benefit assessment process, users with high privacy concerns devote less attention to competing platforms than those with low privacy concerns ([58]). Upon acquiring functional and emotional value from the original platform, they develop stronger relational dependency, manifested through positive WOM and repurchase intentions. Thus, we hypothesize that:

**H9a.** 
*Consumer privacy concerns positively moderate the effect of perceived functional value on post-purchase intentions.*


**H9b.** 
*Consumer privacy concerns positively moderate the effect of perceived emotional value on post-purchase intentions.*


In summary, this study proposes the theoretical model illustrated in Figure 1.

## 4. Research Methodology

### 4.1. Measures

Primary data were collected through an online survey. To ensure conceptual equivalence in the Chinese context, a back-translation procedure was employed. The original English scale items were first translated into Chinese and then independently back-translated by two information systems experts. Subsequently, one professor and three doctoral students in the HCI field were invited to verify the translations. Following their feedback, we adjusted the items for greater semantic consistency and correctness.

The questionnaire comprised two sections. The first section presented measurement items for research constructs. All measurement items were adapted from validated scales in the previous literature, with the original items contextualized to fit the out-of-stock substitution scenario. A 7-point Likert scale was utilized, ranging from “strongly disagree” (1) to “strongly agree” (7). The perceived fit scale was adapted from [101] ([101]), the serendipity items from [105] ([105]), and the personalization measures from [22] ([22]). Perceived functional value was measured using the items derived from [21] ([21]), while items for perceived emotional value were adjusted based on [21] ([21]), and [88] ([88]). Moreover, the measurements of post-purchase intention were derived from [48] ([48]), and the measurements of privacy concerns drew on [90] ([90]). Furthermore, the second section collected demographic information. Following prior research practices, gender, age, education level, income, occupation, and purchase history were included as control variables to mitigate external validity threats. Appendix A lists all the items used for the formal survey.

### 4.2. Data Collection

Data were collected through Credamo, a professional Chinese survey platform, with monetary compensation provided to each participant. To ensure respondents possessed authentic experience with online stockouts and AI-recommended substitutes, an initial screening question was incorporated into the questionnaire. Participants were presented with four stockout recovery options: an apology, monetary compensation, a free gift, or a substitute recommendation. Only those selecting the substitute recommendation were granted access to the main survey, thereby excluding unrelated participants. To further ensure data quality, we enforced multiple screening criteria during sample collection. First, attention-check items were embedded to identify inattentive respondents. Second, leveraging the platform’s reputation system, we established strict participant eligibility criteria: only users with a survey credit score of ≥90% and a historical survey acceptance rate of ≥90% were invited to participate.

Additionally, responses were collected and screened according to the following criteria: (1) exclusion of responses with excessively long or short completion times, (2) removal of patterned responses, and (3) exclusion of participants who failed the attention-check question (‘Please select “1” for this question’). Ultimately, 479 valid responses were retained for analysis. A power analysis conducted using G*Power 3.1 showed that a minimum sample size of 114 was required (effect size f^2^ = 0.15, α = 0.05, power = 0.80). Thus, the sample size of this study satisfied the minimum criteria requirement for Partial Least Squares Structural Equation Modeling (PLS-SEM) analysis. To describe the sample, several demographic factors including gender, age, education level, occupation, income, and online purchase history were collected from respondents. Participant details are presented in Table 1.

### 4.3. Data Analysis and Results

PLS-SEM analysis was conducted using SmartPLS 4. This study integrates multiple theories to predict key target constructs. Prior research has indicated that PLS- SEM is particularly suitable for evaluating causal predictive relationships during theory testing and model development. Given the model’s complexity (7 first-order constructs and 22 indicators), the non-parametric estimation approach of PLS-SEM was employed to minimize the amount of unexplained variance. To identify critical variables required for achieving specific outcomes, the NCA was further employed to determine prerequisites for realizing the expected results.

#### 4.3.1. Measurement Model

Reliability of the measurement model was evaluated using Cronbach’s alpha and composite reliability (CR). As shown in Table 2, the Cronbach’s alpha and CR of each construct are greater than 0.7. Therefore, all constructs in the study have good internal consistency and stability. The standardized factor loadings of each indicator are greater than 0.7, and the average variance extracted (AVE) is greater than 0.5. The measurement model of this study has good convergent validity. The discriminant validity of the model was assessed using Fornell-Larcker’s criterion and heterotrait-monotrait ratio (HTMT). As Table 3 and Table 4 show, the square root of AVE for each construct was greater than its correlations with other latent variables, and the HTMT ratio values are below the threshold of 0.9, indicating that the research model has good discriminant validity.

Due to the self-reported and single-source nature of the data, common method bias (CMB) was examined. First, collinearity was assessed using variance inflation factors (VIF), with all values ranging from 1.327 to 2.690, which were below the threshold of 3.3, indicating that this study had no multicollinearity issue. Second, the unmeasured latent method construct (ULMC) was used in AMOS to analyze the influence of common method variation. Results showed minimal changes in model fit indices after adding the common method factor (ΔRMSEA = 0.006, ΔTLI = 0.007, ΔCFI = 0.007), suggesting no significant improvement in fit. Additionally, we incorporated a common method factor into PLS model proposed by [60] ([60]). As shown in Table 5, substantive factor explained 78.2% of the variance, while method factors accounted for only 0.8%. To sum up, there was no significant common method bias in this study.

#### 4.3.2. Structural Model

Following the suggestion of [33] ([33]), bootstrapping (a procedure with 5000 samples) was conducted to analyze the conceptual model. Path coefficients, effect size (f^2^), and R^2^ values were used to examine the explanatory power, whereas Q^2^ values were employed to assess the model’s predictive relevance. As summarized in Table 6, the R^2^ values for all reflective constructs exceeded the threshold of 0.1, indicating that the model explained adequate variance in the endogenous constructs. According to [19] ([19]), the *p*-value alone cannot reveal the effect size of predictor variables. Consequently, f^2^ values were calculated to estimate the effect sizes of the exogenous variables on the endogenous variables. As indicated in Table 6, all Q^2^ values were greater than 0, suggesting that the exogenous variables exhibit predictive relevance for the endogenous variables. Moreover, as Table 6 shows, path coefficient analysis revealed that perceived fit, personalization, and serendipity positively influenced perceived functional value, perceived emotional value, and post-purchase intentions. Thus, Hypotheses 1–7 were supported.

As presented in Figure 2, perceived functional value is positively influenced by perceived fit (*β* = 0.356, *p* < 0.001), perceived personalization (*β* = 0.183, *p* < 0.001), and perceived serendipity (*β* = 0.323, *p* < 0.001). Similarly, perceived emotional value is positively affected by perceived fit (*β* = 0.253, *p* < 0.01), perceived personalization (*β* = 0.394, *p* < 0.001), and perceived serendipity (*β* = 0.270, *p* < 0.001). Perceived functional value (*β* = 0.314, *p* < 0.001) and perceived emotional value (*β* = 0.396, *p* < 0.001) were both significant drivers of post-purchase behavioral intention. Thus, hypotheses H1 to H7 are supported.

To examine the differential effects of perceived fit, personalization, and serendipity on both functional and emotional value, we analyzed standardized path coefficients, effect sizes “f^2^”, and the path comparison approach outlined by ([17]). [17] ([17]) proposed calculating percentile *p*-values by comparing path coefficients across 5000 bootstrap samples to assess the existence of differential effects. The results showed that the perceived fit had a larger effect on perceived functional value (f^2^ = 0.153) than on perceived emotional value (f^2^ = 0.095), the bootstrap samples also revealed that perceived fit had a stronger effect on perceived functional value than on perceived emotional value (*p* = 0.028). These results supported H8a. Moreover, the effect sizes indicated that perceived personalization had a larger effect on perceived emotional value (f^2^ = 0.179) compared to perceived functional value (f^2^ = 0.031). Additionally, the bootstrap samples also indicated that perceived personalization had a stronger effect on perceived emotional value (f^2^ = 0.179) compared to perceived functional value, thus supporting H8b. Additionally, perceived serendipity showed no significant difference in its effect on perceived emotional value compared to perceived functional value (*p* = 0.168). Thus, H8c was supported.

We further analyzed the moderating effect of privacy concerns on the relationship between perceived value and post-purchase intention. Results indicated that privacy concerns positively moderated the influence of perceived functional value on post-purchase intention (*β* = 0.149, *p* = 0.011). Thus, H9a was supported, suggesting that the positive effect of perceived functional value on post-purchase intentions is stronger among consumers with high privacy concerns. Specifically, when AI-recommended substitutes demonstrate high functional value (e.g., practicality), these consumers exhibit significantly higher levels of repurchase intention and positive WOM than their low-concern counterparts.

However, no significant moderating effect was found in the relationship between perceived emotional value and post-purchase intention (*β* = -0.048, *p* = 0.378). H9b was not supported, indicating no significant difference in the impact of perceived emotional value on post-purchase intentions between consumers with high versus low privacy concerns. Furthermore, Figure 3 was plotted to illustrate the moderating effects at one standard deviation above and below the mean of privacy concerns, showing stronger marginal increases in functional value’s behavioral impact under high privacy concern conditions.

#### 4.3.3. Robustness Checks

To validate the reliability of the main findings, this study conducted a series of robustness tests. First, the finite mixture PLS (FIMIX-PLS) approach was reapplied to examine potential unobserved heterogeneity bias. Given a minimum sample size requirement of 114, a maximum of 4 segments could be extracted from the 479 participants, provided the minimum sample size for reliable parameter estimation was met ([61]). Since the minimum proportion of the fourth segment was 0.109, the three-segment solution (K = 3) was considered the optimal segmentation. Default settings in FIMIX-PLS were employed, including a stop criterion of 1.0E-10, a maximum of 5000 iterations, and the number of repetitions of 10. As shown in Table 7, the results suggested AIC_3_ (modified AIC with Factor 3) pointed to a 3-segment solution, while CAIC (consistent AIC) indicated a 2-segment solution. All different criteria did not converge on the same number of segments, and the EN (normed entropy statistic) values were below 0.50, indicating that unobserved heterogeneity does not pose a threat to our results.

Furthermore, we examined potential endogeneity bias, which may stem from sample selection bias, omitted variable bias, or reverse causality ([94]). Following the methodological approach of [82] ([82]), this study employed the Heckman two-stage estimation procedure using Stata 17.0 software ([35]). As shown in Table 8, the regression results demonstrate that the main findings remain statistically robust. Finally, we tested for nonlinear effects to ensure that the relationships between variables in the model conform to linear assumptions. By introducing quadratic terms to construct a polynomial model, the results fully support that the relationships between variables in the model are linear (*p* > 0.05), suggesting that there are no quadratic effects (e.g., U-shaped or S-shaped) among the variables (see Table 9).

### 4.4. NCA Results

Although SEM examines correlations among variables, NCA serves as a valuable data analysis method rooted in necessity logic. It identifies essential prerequisites for outcomes and effectively complements the sufficiency-based PLS-SEM approach. Thus, the integration of PLS-SEM and NCA supports a more comprehensive and symmetrical analysis of the research model. In this study, NCA was employed to identify necessary prerequisites for perceived functional value, perceived emotional value, and post-purchase intentions. Permutation tests with 10,000 random samples were conducted to assess the necessity of the antecedents, and the ceiling line was estimated using the Ceiling Envelopment-Free Disposal Hull (CE-FDH). According to [25] ([25]), the requisite condition’s effect size should exceed 0.1 with *p*-values below 0.05 to hold practical significance. As shown in Table 10 and Figure 4, perceived fit emerged as a necessary condition for perceived functional value, while perceived fit, perceived personalization, and perceived serendipity were necessary conditions for perceived emotional value. Both perceived functional value and perceived emotional value were identified as necessary prerequisites for post-purchase intentions.

The critical test was conducted using a bottleneck table, which shows the essential thresholds of the conditions required for a specific level of the outcome. Results in Table 11 indicated that to achieve high levels of post-purchase intentions (90%), perceived functional value and perceived emotional value should reach minimum levels of 68.7% and 64.7%. To achieve a high level of perceived functional value (90%), a minimum of 55.6% perceived fit was required. For high level (90%) of perceived emotional value, at least 66.7% perceived fit, 47.1% perceived serendipity, and 66.7% perceived personalization were necessary.

## 5. Discussion and Implications

### 5.1. Main Findings

Despite the exponential growth of AI algorithm recommendations across industries, their economic impact and underlying mechanisms in out-of-stock substitution scenarios remain inadequately explored. Drawing on the Affordance Theory and Perceived Value Theory, this study examines how the content affordances of AI recommendations shape customer value perceptions and post-purchase intentions. PLS-SEM and NCA methods are integrated to examine both sufficient and necessary conditions, thereby enhancing causal understanding of structural relationships within the proposed framework.

First, the PLS-SEM results demonstrate that the content affordances of perceived fit, perceived personalization, and perceived serendipity are key antecedents of consumer perceived value. Specifically, the feature alignment between a substitute and the preferred product reflects a deep-level task-technology fit. Perceived personalization fosters self-congruity with the substitute, while perceived serendipity introduces valuable discovery by offering unexpected alternatives that are consistent with consumers’ latent preferences. Collectively, these affordances give rise to both perceived functional and emotional value. Furthermore, this study reveals the differential effects of technological characteristics on distinct consumer value dimensions. Perceived fit has a stronger impact on consumers’ perceived functional value, whereas perceived personalization exerts a greater influence on perceived emotional value. In contrast, perceived serendipity exhibits no significant differential effects between functional and emotional values. These findings validate the perspectives proposed by [3] ([3]) and [36] ([36]): utilitarian elements of applications or products prompt consumers to evaluate instrumental benefits, while hedonic attributes exert a more pronounced influence on affective responses than cognitive assessments.

Second, this study employs post-purchase intentions to assess the long-term efficacy of service recovery. The results demonstrate that consumers’ perceived functional and emotional values derived from AI-recommended substitutes predict behavioral intentions such as repeat purchases and positive WOM. Previous studies have revealed the positive effects of consumers’ perceived value on repurchase, recommendation, and continuance intention in various contexts including hospitality services, chatbots, online gaming, and VR services. Our results are in line with these findings. When users perceive utilitarian benefits from alternative assortments or experience hedonic enjoyment, they develop heightened behavioral loyalty and affective commitment toward the original platform.

Third, the results reveal that privacy concerns positively moderate the relationship between perceived functional value and consumers’ post-purchase intentions. Privacy- sensitive users exhibit reinforced loyalty to the original platform following an effective functional recovery. In contrast, privacy concerns do not significantly moderate the relationship between perceived emotional value and purchase decisions. Thus, H9b is not supported. This may be explained through two distinct value pathways. In the functional value pathway, consumers’ perception of substantive utilitarian benefits activates cost–benefit calculations. Users with high privacy concerns, given their risk-averse nature, may strengthen their dependency on the original platform, resulting in enhanced loyalty and repurchase intentions. Within the emotional value pathway, high levels of emotional value (e.g., surprise, feeling understood) effectively alleviate negative emotions triggered by stockouts, serving as a powerful form of affective compensation and emotional support, which reinforces users’ affective trust and relational commitment toward the platform ([108]). According to social exchange theory, a long-term reciprocal and affect-laden relational paradigm motivates users to maintain stable existing relationships ([13]), leading to greater tolerance toward temporary platform failures (e.g., stockouts), lower switching intentions, and reduced perception of platform opportunism ([28]). Consequently, the moderating effect of privacy concerns is rendered insignificant.

In addition, NCA demonstrates that both perceived functional value and emotional value are necessary conditions for post-purchase intentions. The analysis shows that AI- recommended substitute sets must be efficient, useful, and practical to deliver functional benefits, while simultaneously evoking positive affective experiences that drive repeat purchases and positive WOM. Moreover, perceived fit constitutes a key determinant for functional value, making the offering of similar substitutes during stockouts essential. Meanwhile, perceived fit, personalization, and serendipity are critical determinants of perceived emotional value, which requires that recommended substitutes should match consumer preferences, fulfill expectations, and provide unexpected valuable information to enhance consumers’ emotional value perceptions.

### 5.2. Theoretical Implications

Our study makes a significant original contribution to the literature on substitute recovery in out-of-stock scenarios. First, although AI recommender systems are widely implemented in stockout contexts, few studies have examined how AI-recommended substitute sets yield economic benefits. Moreover, prior studies have primarily focused on the short-term impacts of conventional recovery measures (e.g., product upgrades, apologies, and compensation) from perspectives like contextual effects, stockout cues, and product attributes. In contrast, this study validates AI-recommended substitutes as a potent service recovery mechanism. By identifying three crucial content features of AI-recommended substitutes—perceived fit, perceived personalization, and perceived serendipity—our findings reveal the pivotal role of AI substitution strategies in shaping long-term customer relationships (e.g., repeat purchases and positive WOM), thereby extending the theoretical perspectives of stockout recovery and technology affordance.

Second, this study examines the underlying mechanisms through which content affordances influence consumers’ post-purchase intentions. Previous service recovery frameworks have emphasized the critical role of fairness and justice ([91]), but their explanatory power is limited in AI-driven contexts ([59]). This study introduces perceived value as a pivotal mediator to construct a dual-pathway model. Grounded in the premise that consumers’ attitudes toward AI technology determine the successful delivery of a company’s value propositions ([55]), our findings reveal that AI-recommended substitutes serve as an effective economic recovery strategy, wherein content affordances simultaneously trigger perceptions of functional value and emotional value, thereby directly driving post-recovery repurchase intentions and WOM. Furthermore, the results show the differential effects of content affordance dimensions on two value pathways: perceived fit primarily shapes functional value, whereas perceived personalization predominantly influences emotional value, indicating that the alignment among AI recovery strategies, stockout incidents, and individual needs significantly enhances perceived value (e.g., “task-technology fit” and “self-technology fit”). The results advance the theoretical framework of AI-driven service recovery and deepen the understanding of value formation mechanisms in human-AI interaction.

Third, our study reveals that privacy concerns positively moderate the relationship between perceived functional value and consumers’ post-purchase intentions. Previous research has suggested that privacy concerns represent a negative technology affordance that diminishes outcome valences ([79]), undermining consumers’ adoption intention, continuance usage, and recommendation intention ([52]). However, existing research has not explored platform switching under service failure contexts, particularly when users derive significant functional value from a platform’s technological support. From the lens of social exchange theory, we identify a positive moderating effect of privacy concerns. Specifically, consumers’ perceived utilitarian value in service recovery emerges as a key prerequisite for accepting privacy costs. When users with higher privacy concerns perceive functional benefits from the original platform’s service recovery, a rational cost–benefit calculus motivates them to maintain the existing relationship, manifesting as stronger loyalty and repurchase intentions. Research on switching costs also confirms that customers’ loss aversion positively affects retention, advancing theories of privacy concerns and disclosure ([45]).

### 5.3. Managerial Implications

This study provides a solid theoretical and methodological foundation to guide platform operators in designing recommendation algorithms and strategies for out-of-stock substitution. First, it validates AI-recommended substitutes as an effective proactive service recovery strategy. Although most platforms leverage AI recommender systems to mitigate the negative impacts of stockouts, significant variation exists in the design of their choice sets’ content features, particularly in balancing relevance and diversity/serendipity. Our findings indicate that alternative sets integrating fit, personalization, and serendipity help build stronger long-term customer relationships. Specifically, algorithms should recommend substitutes with high functional alignment to meet rational problem-solving needs, while also providing personalized and serendipitous options to evoke emotional value. According to the service recovery paradox, a successful recovery from service failures can yield higher consumer satisfaction levels than before the incident occurred.

Second, the results indicate that perceived serendipity is a necessary condition for consumers’ value perception, with unexpectedly delightful recommendations for driving service recovery effectiveness. Therefore, by proactively identifying and responding to latent consumer preferences through unexpectedly fitting stockout alternatives, platforms can strengthen value perception and service identification, thereby cultivating long-term loyalty. However, the potential risks of “excessive serendipity” must be acknowledged. For instance, over-recommending alternatives that deviate from users’ explicit preferences may undermine the perceived reliability of the AI -driven system and breed skepticism about its intentions. Moreover, excessive serendipity can induce cognitive conflict and choice overload, potentially leading to purchase abandonment ([101]). Therefore, platform managers should strive to balance recommendation accuracy with exploratory shopping, first ensuring that substitutes satisfy core functional needs before introducing moderately surprising options.

Third, this study reveals individual heterogeneity in privacy concerns during AI- driven service recovery. Privacy concerns positively moderate the effect of perceived functional value on post-purchase intentions. Notably, consumers with higher privacy sensitivity demonstrate stronger loyalty to the original platform following an effective recovery. Consequently, platforms should increase investments in AI infrastructure and algorithmic R&D to enhance the quality and utility of recommendations. By delivering superior value that outweighs the costs of consumers’ privacy disclosure, platforms can convert highly privacy-conscious users into their most stable customer base. Moreover, platforms should establish a transparent and controllable data environment. For instance, they should clearly explain to consumers the logic of data collection, usage, and protection, as well as provide privacy protection tools that enhance user data access and control. Ultimately, these approaches cultivate the privacy literacy and empowerment among consumers to internalize privacy concerns as a switching cost, thereby strengthening competitive barriers and ensuring sustainable customer loyalty.

## 6. Limitations and Future Research

This study has several limitations that suggest potential avenues for future research. First, AI recommender systems possess multiple dimensions of technology affordance; this study focused solely on the impacts of content affordance and privacy concerns on consumers’ post-purchase intentions in out-of-stock scenarios. Previous research has confirmed that technical features such as the explainability, interactivity, and information presentation of AI recommender systems influence an individual’s understanding of the recommendation logic. The perceived transparency and algorithmic fairness may shape consumers’ value perception and subsequent behaviors. Therefore, future studies should explore other technical dimensions or adopt an integrated framework to evaluate the efficacy of AI-enabled service recovery. Additionally, situational cues, including the functional or hedonic attributes of products, product involvement, and causes of stockouts, also influence consumers’ processing of stock-out information. Future research should also investigate the boundary conditions for the efficacy of AI-driven substitution recovery strategies.

Second, this study establishes a dual-pathway recovery framework focusing on the perspective of perceived value. Future work should incorporate diverse theoretical lenses to further investigate the psychological mechanisms underlying consumer responses to AI-driven recovery. For instance, it could explore whether emotional value fosters WOM through affective trust, or whether the efficacy of functional value is contingent upon perceived fairness. Such efforts would contribute to constructing a more comprehensive and nuanced causal pathway from AI service features to user behavior.

Third, the reliance on self-reported data in this study may constrain the external validity of the main findings. Future research could employ laboratory experiments, field experiments, or analyses of real-world behavioral data to strengthen external validity. Furthermore, since our research was conducted in China, cultural factors pertaining to privacy and relationships may limit the generalizability of the model. Cross-cultural comparisons and extensions to diverse contexts are needed to validate these findings and enhance their robustness.

## Figures and Tables

**Figure 1 behavsci-15-01507-f001:**
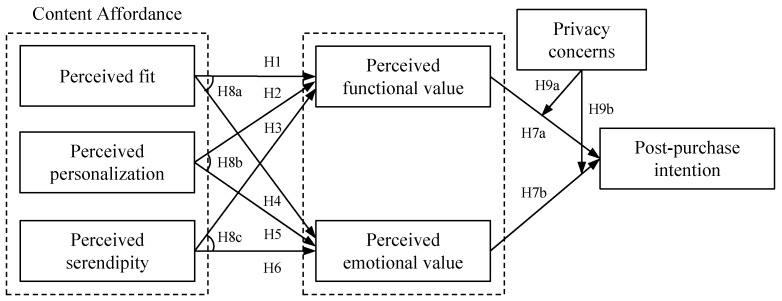
Research model.

**Figure 2 behavsci-15-01507-f002:**
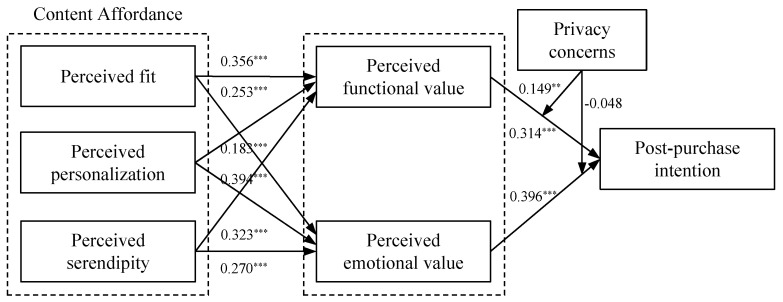
Structural model results. Note: ***, ** statistically significant at the 1 percent and 5 percent levels.

**Figure 3 behavsci-15-01507-f003:**
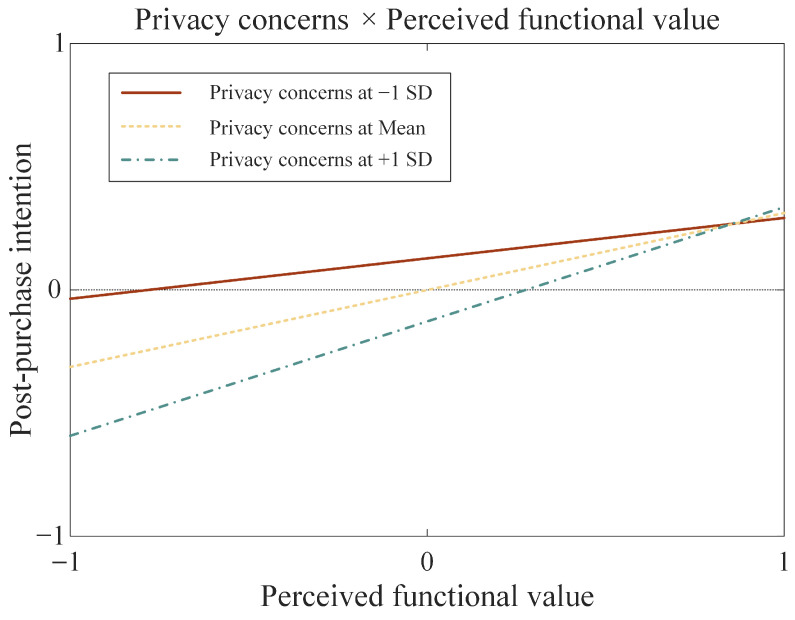
Moderating effect of PC between PFV and PPI.

**Figure 4 behavsci-15-01507-f004:**
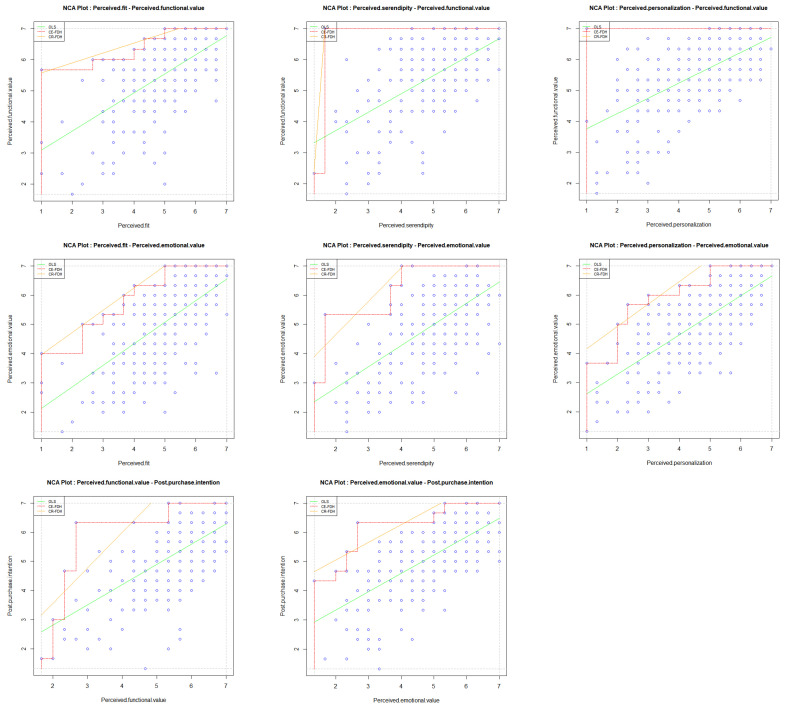
NCA scatter plots. Note: blue dots represent observations.

**Table 1 behavsci-15-01507-t001:** Demographics of respondents.

	Variables	Numbers	Percentage (%)
Gender	Male	211	44.1
	Female	268	55.9
Age	Under 20	20	4.2
	21–30	249	52.0
	31–40	171	35.7
	41–50	25	5.2
	50 and above	14	2.9
Education	Junior college and below	42	8.8
	Undergraduate	357	74.5
	Postgraduate	80	16.7
Occupation	Student	90	18.8
	Employee	383	79.9
	Others	6	1.3
Monthly income (RMB)	Under 2000	62	12.9
	2000–3999	31	6.5
	4000–5999	56	11.7
	6000–7999	75	15.7
	8000–9999	102	21.3
	10,000 and above	153	31.9
Purchase history (Year)	Below 1	14	2.9
	2–3	93	19.4
	4–5	123	25.7
	6–7	104	21.7
	8 and above	145	30.3

**Table 2 behavsci-15-01507-t002:** Reliability and validity analysis.

Variable	Item	Loading	Cronbach’s α	CR	AVE
Perceived fit	PF1	0.834	0.794	0.879	0.708
(PF)	PF2	0.855			
	PF3	0.834			
Perceived personalization	PP1	0.869	0.859	0.914	0.780
(PP)	PP2	0.898			
	PP3	0.882			
Perceived serendipity	PS1	0.856	0.769	0.866	0.683
(PS)	PS2	0.755			
	PS3	0.864			
Perceived functional value	PFV1	0.895	0.859	0.914	0.780
(PFV)	PFV2	0.877			
	PFV3	0.877			
Perceived emotional value	PEV1	0.885	0.887	0.93	0.815
(PEV)	PEV2	0.910			
	PEV3	0.913			
Post-purchase intention	PPI1	0.879	0.848	0.908	0.767
(PPI)	PPI2	0.852			
	PPI3	0.897			
Privacy concerns	PC1	0.945	0.962	0.972	0.897
(PC)	PC2	0.944			
	PC3	0.953			
	PC4	0.946			

**Table 3 behavsci-15-01507-t003:** Discriminant validity analysis.

	PP	PPI	PFV	PEV	PS	PF	PC
PP	0.883						
PPI	0.579	0.876					
PFV	0.626	0.670	0.883				
PEV	0.735	0.702	0.750	0.903			
PS	0.641	0.538	0.615	0.647	0.827		
PF	0.663	0.551	0.636	0.647	0.490	0.841	
PC	−0.382	−0.359	−0.254	−0.424	−0.355	−0.307	0.947

**Table 4 behavsci-15-01507-t004:** Heterotrait–Monotrait (HTMT) ratio.

	PP	PPI	PFV	PEV	PS	PF	PC
PP							
PPI	0.678						
PFV	0.729	0.783					
PEV	0.841	0.809	0.858				
PS	0.78	0.656	0.75	0.771			
PF	0.801	0.671	0.769	0.767	0.621		
PC	0.42	0.397	0.279	0.458	0.402	0.351	

**Table 5 behavsci-15-01507-t005:** Common method bias analysis.

Construct	Indicator	Substantive Factor Loading (R1)	R1^2^	Method Factor Loading (R2)	R2^2^
Perceived fit	PF1	0.840	0.706	−0.068	0.005
	PF2	0.848	0.719	0.037	0.001
	PF3	0.836	0.699	0.030	0.001
Perceived personalization	PP1	0.868	0.753	0.096	0.009
	PP2	0.901	0.812	−0.084	0.007
	PP3	0.881	0.776	−0.010	0.000
Perceived serendipity	PS1	0.854	0.729	0.036	0.001
	PS2	0.777	0.604	−0.184	0.034
	PS3	0.849	0.721	0.124	0.015
Perceived functional value	PFV1	0.894	0.799	0.006	0.000
	PFV2	0.875	0.766	0.010	0.000
	PFV3	0.881	0.776	−0.016	0.000
Perceived emotional value	PEV1	0.881	0.776	0.190	0.036
	PEV2	0.914	0.835	−0.196	0.038
	PEV3	0.914	0.835	0.007	0.000
Post-purchase intention	PPI1	0.881	0.776	−0.046	0.002
	PPI2	0.848	0.719	0.111	0.012
	PPI3	0.898	0.806	−0.061	0.004
Privacy concerns	PC1	0.946	0.895	0.028	0.001
	PC2	0.941	0.885	−0.030	0.001
	PC3	0.953	0.908	−0.013	0.000
	PC4	0.948	0.899	0.014	0.000
	Average		0.782		0.008

**Table 6 behavsci-15-01507-t006:** PLS structural model results.

Path	Std. Beta	*t*-Value	2.5%CI	97.5%CI	R^2^	Q^2^	f^2^
H1: PF→PFV	0.356	6.702	0.250	0.459			0.153
H2: PP→PFV	0.183	3.148	0.069	0.297	0.540	0.414	0.031
H3: PS→PFV	0.323	5.958	0.217	0.427			0.132
H4:PF→PEV	0.253	5.736	0.168	0.340			0.095
H5: PP→PEV	0.394	8.549	0.303	0.481	0.628	0.505	0.179
H6: PS→PEV	0.270	6.159	0.182	0.356			0.114
H7a: PFV→PPI	0.314	5.336	0.202	0.434	0.563	0.419	0.088
H7b: PEV→PPI	0.396	7.614	0.293	0.499			0.129

**Table 7 behavsci-15-01507-t007:** Assessment of unobserved heterogeneity.

	Number of Segments		
Criteria	K = 1	K = 2	K = 3
AIC (Akaike’s information criterion)	2867.876	2728.44	2684.011
AIC_3_ (modified AIC with Factor 3)	2881.876	2757.44	2728.011
AIC_4_ (modified AIC with Factor 4)	2895.876	2786.44	2772.011
BIC (Bayesian information criterion)	2926.28	2849.419	2867.566
CAIC (consistent AIC)	2940.28	2878.419	2911.566
HQ (Hannan-Quinn criterion)	2890.835	2775.998	2756.169
MDL_5_ (minimum description length with factor 5)	3271.895	3565.336	3953.785
LnL (LogLikelihood)	−1419.938	−1335.22	−1298.006
EN (normed entropy statistic)	NA	0.353	0.447
NFI (non-fuzzy index)	NA	0.406	0.446
NEC (normalized entropy criterion)	NA	309.969	264.851

Note: NA: not available.

**Table 8 behavsci-15-01507-t008:** Assessment of endogeneity bias.

Test	*β*	*p*-Value	z	Conclusion
PF→PFV (Selection DV = PEV; IV = PP)	0.289	0.000 ***	5.97 ***	No bias present
PF→PEV (Selection DV = PFV; IV = PP)	0.467	0.000 ***	5.87 ***	No bias present
PP→PFV (Selection DV = PEV; IV = PF)	0.154	0.006 ***	2.76 ***	No bias present
PP→PEV (Selection DV = PFV; IV = PF)	0.505	0.000 ***	13.01 ***	No bias present
PS→PFV (Selection DV = PEV; IV = PP)	0.127	0.023 **	2.27 **	No bias present
PS→PEV (Selection DV = PFV; IV = PP)	0.444	0.000 ***	7.94 ***	No bias present
PFV→PPI (Selection DV = PC; IV = PEV)	0.585	0.000 ***	9.04 ***	No bias present
PEV→PPI (Selection DV = PC; IV = PFV)	1.043	0.000 ***	4.58 ***	No bias present

Note: DV = dependent variables, IV = independent variables; ***, ** statistically significant at the 1 percent and 5 percent levels.

**Table 9 behavsci-15-01507-t009:** Assessment of nonlinear effects.

Nonlinear Relationship	*β*	*p*-Value
QE(PF)→PFV	0.029	0.295
QE(PF)→PEV	0.037	0.068
QE(PP)→PFV	−0.023	0.509
QE(PP)→PEV	−0.039	0.110
QE(PS)→PFV	−0.052	0.232
QE(PS)→PEV	0.027	0.332
QE(PFV)→PPI	0.007	0.835
QE(PEV)→PPI	0.010	0.780

Note: QE: quadratic effect.

**Table 10 behavsci-15-01507-t010:** NCA effect sizes.

	PFV		PEV		PPI	
	CE-FDH	*p*-Value	CE-FDH	*p*-Value	CE-FDH	*p*-Value
	Effect Size		Effect Size		Effect Size	
PF	0.100	0.000	0.178	0.000		
PP	0	1	0.154	0.000		
PS	0.026	0.590	0.132	0.006		
PFV					0.201	0.000
PEV					0.143	0.001

**Table 11 behavsci-15-01507-t011:** Bottleneck table (percentages).

PFV		PEV				PPI		
Y (%)	PF	Y (%)	PF	PS	PP	Y (%)	PFV	PEV
NN	NN	NN	NN	NN	NN	NN	NN	NN
10	NN	10	NN	NN	NN	10	6.2	NN
20	NN	20	NN	NN	NN	20	6.2	NN
30	NN	30	NN	5.9	NN	30	12.5	NN
40	NN	40	NN	5.9	16.7	40	12.5	NN
50	NN	50	22.2	5.9	16.7	50	12.5	NN
60	NN	60	22.2	5.9	16.7	60	18.8	17.6
70	NN	70	33.3	5.9	22.2	70	18.8	17.6
80	27.8	80	44.4	41.2	33.3	80	18.8	23.5
90	55.6	90	66.7	47.1	66.7	90	68.7	64.7
100	66.7	100	66.7	47.1	66.7	100	68.7	70.6

Note: NN represent “Not Necessary”.

## Data Availability

The data presented in this study are available on request from the corresponding author. The data are not publicly available due to privacy or ethical restrictions.

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
