# Peer review of "The Impact of AI-Recommended Content Affordances on Post-Purchase Intention in Stockout Substitution Scenarios"

_behavsci, 2025, doi:10.3390/bs15111507_

Round 1

Reviewer 1 Report

Comments and Suggestions for Authors

The manuscript presents a timely and relevant study examining the role of AI recommender systems in stockout substitution contexts. Overall, it demonstrates a solid methodological foundation and theoretical grounding, and it has the potential to make a meaningful contribution to the literature on service recovery and digital consumer behavior. However, several conceptual and theoretical aspects require further clarification and development before the paper can be considered.

The conceptual boundaries among the three affordance dimensions that perceived fit, personalization, and serendipity, are not fully distinct. Their empirical separation is statistically supported, yet theoretically they overlap considerably. This may raise concerns about redundancy or construct clarity.

Moreover, the research model itself is relatively linear, focusing solely on post-purchase intention as the outcome variable. Incorporating mediating or moderating constructs, such as trust, satisfaction, or perceived justice, could deepen the explanatory power of the framework and align the study more closely with the complex mechanisms typical of high-impact behavioral research.

The interpretation of the moderating effect of privacy concerns also requires further theoretical grounding. The finding that privacy concern strengthens the relationship between perceived functional value and post-purchase intention, but not between emotional value and intention, is intriguing but insufficiently explained. The discussion currently describes this as an empirical observation rather than unpacking the psychological or behavioral mechanisms underlying it. A richer theoretical rationale—perhaps drawing on social exchange theory or priva

Another issue concerns external validity. Since the data were collected exclusively from Chinese consumers through the Credamo platform, the findings may be influenced by cultural factors such as privacy perception. Acknowledging this limitation more explicitly and suggesting cross-cultural or experimental replication for future research would strengthen the paper’s generalizability and scholarly impact.

In summary, I consider the paper promising but recommend minor revision before it can be deemed suitable for publication.

Reviewer 2 Report

Comments and Suggestions for Authors

This paper offers a timely and relevant contribution to behavioural science and information systems by examining how AI recommender system affordances—fit, personalisation, and serendipity—influence post-purchase intentions in stockout recovery contexts. It effectively integrates Technology Affordance Theory and Perceived Value Theory using PLS-SEM and NCA, resulting in a conceptually solid and methodologically rigorous analysis.

The study shows strong potential for publication after moderate theoretical refinement and clearer methodological exposition. To strengthen the paper before acceptance, the authors should:

  • Clarify theoretical novelty—explain how the framework extends prior AI affordance and stockout recovery studies.

  • Enhance methodological transparency, detailing sampling, translation, and context (e.g., the Credamo process, participant profile, and stockout examples).

  • Condense and structure the introduction and hypotheses for clarity, using subheadings or diagrams (e.g., “Functional vs. Emotional Pathways”).

  • Deepen discussion of privacy moderation, cultural bias, and international generalisability.

The paper’s strengths lie in addressing a clear research gap on AI-driven service recovery, offering an original intersection of personalization and loyalty, and presenting nuanced findings on privacy as a positive moderator. However, improvements are needed to articulate its theoretical innovation, broaden cross-disciplinary significance beyond consumer psychology, and clarify methodological rigor.

Further refinements should include providing a complete structural model figure with path coefficients, interpreting the non-significant moderation (H9b) with theoretical reasoning, and more critically discussing the study’s limitations and future research directions—especially around AI transparency, cross-platform loyalty, and cultural applicability. With these revisions, the paper will make a strong and lasting contribution to the journal’s quality and relevance.

Reviewer 3 Report

Comments and Suggestions for Authors

I would like to thank the authors for their careful and well-executed work on this relevant topic. The manuscript presents a coherent analysis of how AI-recommended content affordances influence post-purchase intention in stockout substitution scenarios. The integration of PLS-SEM and Necessary Condition Analysis adds methodological strength and novelty to the study. The paper is well structured, and the argument develops logically. However, the theoretical section could be more focused, engaging more directly with debates on privacy calculus, justice in service recovery, dual-process models of perceived value, and explainable AI. These additions would broaden the paper’s theoretical contribution and international relevance. APA referencing should also be reviewed for consistency. The methodological design is solid, but further robustness could be achieved by reporting confidence intervals, testing for measurement invariance, and addressing possible endogeneity or sample bias. The results are clearly presented and aligned with the hypotheses. It would be helpful to clarify the link between PLS (sufficiency) and NCA (necessity) and to reflect briefly on the potential downsides of excessive serendipity. The discussion and implications are relevant for both theory and practice. Future studies could adopt experimental or field designs and explore different market contexts. This is a rigorous and timely manuscript with clear potential for publication once minor theoretical and methodological refinements are addressed. Best wishes to the authors for their continued research on this important topic.

Reviewer 4 Report

Comments and Suggestions for Authors

The manuscript is a valuable scientific contribution that brings original insights into the field of consumer behavior research and the use of AI-based recommendation systems. The authors clearly identify gaps in existing research, but at the same time fill them in a targeted manner through their own empirical research. The study provides new theoretical and practical insights that can be beneficial to both the academic community and practitioners in the field of online retail and digital marketing.

Methodologically, the study is robust, using bootstrapping, common method bias control, questionnaire pretesting, and thorough verification of the reliability and validity of measurement tools using Cronbach's alpha, composite reliability (CR), AVE, and HTMT criteria, thereby increasing the credibility and stability of the results. 

Overall, the article successfully integrates theoretical frameworks with rigorous empirical analysis, offering a comprehensive understanding of how AI-generated substitute recommendations influence consumer perceptions and behavior. Its findings not only advance academic knowledge in the domains of consumer behavior, service recovery, and AI technology adoption, but also provide actionable guidance for online retailers seeking to optimize substitute offerings.

The authors also provide practical recommendations for online platforms, emphasizing the integration of fit, personalization, and serendipity when designing AI-generated substitute offerings.

The study sets a solid foundation for future research and represents a meaningful contribution to both theory and practice.

Author Response

Dear Reviewer,

Thank you very much for your time and for the positive and encouraging feedback on our manuscript. We are sincerely grateful for your supportive assessment.

We appreciate your recognition of our attempt to bridge the research gaps in this area and are reassured that you found the methodological approach to be robust. Your comments regarding the theoretical and practical value of our work are particularly motivating.

In response to your overall endorsement, we have made further revisions to the manuscript and double-checked the formatting to ensure it aligns with the journal's standards, striving to improve the overall clarity and presentation.

Thank you once again for your thoughtful and constructive review. We look forward to the possibility of contributing to the journal.

Sincerely,

The Authors

Round 2

Reviewer 2 Report

Comments and Suggestions for Authors

Both conceptual framing and methodological precision have significantly improved in this updated version. The integration of Technology Affordance Theory and Perceived Value Theory is well-articulated, and the empirical results are clearly presented with robust analysis (PLS-SEM and NCA). However, further refinement of the discussion is recommended to more succinctly highlight the theoretical novelty and managerial implications. Some paragraphs could be condensed to improve readability and flow. Also, could you make sure consistent citation formatting and correct minor grammatical errors? Overall, the paper provides strong contributions to AI-driven service recovery and consumer behavior literature and is now close to publishable quality
